# Prevalence and risk factors associated with intestinal parasitosis in children with diarrhoea in the Huambo municipality

Ednogildo Domingos Miguel Sachocal[1]☯, Sandra Cristina Ribeiro Buta da Costa[2]☯, Hermenegildo Osvaldo Chitumba [3,4]☯ *

**1** Department of Medicine, School of Medicine, José Eduardo dos Santos University, Huambo, Angola, **2** Paediatrics Department, Huambo General Hospital, Huambo, Angola, **3** Department of Pathology and Diagnostic Methods, School of Medicine, José Eduardo dos Santos University, Huambo, Angola, **4** Global Health and Tropical Medicine (GHTM), Instituto de Higiene e Medicina Tropical (IHMT), Universidade Nova de Lisboa, Lisbon, Portugal

☯ These authors contributed equally to this work.
* Chitumba16@gmail.com

## Abstract

Intestinal parasitic infections are classified as neglected tropical diseases and represent a serious public health issue, particularly in developing countries, where they often manifest as diarrhoeal syndromes. A prospective, cross-sectional, observational study was conducted with the aim of describing the prevalence of intestinal parasites in children aged 9 months to 14 years who were hospitalised with a diagnosis of diarrhoeal disease in the paediatric department of Huambo General Hospital. Of the 173 stool samples analysed, 47 (27.2%) were positive, with a predominance of helminths (22.5%), among which *Necator americanus* (46.81%) and *Strongyloides stercoralis* (46.81%) were the most prevalent. The only protozoan identified was *Giardia lamblia* (12.77%) and *Taenia spp.* (4.26%). Although found in low percentages, this study demonstrated the presence of three groups of parasites infecting children, with helminths being the most predominant, followed by protozoa and cestodes. Additionally, the variables studied (age, sex, origin, education level, parents' occupation, source of drinking water, hygiene practices, and open defecation) did not constitute risk factors for the prevalence of intestinal parasitic infections.

## Author summary

Intestinal parasitic infections are a major public health concern in developing countries, particularly among young children who are vulnerable to diarrhoeal diseases. In this study, we investigated the prevalence and risk factors of intestinal parasitosis in children with diarrhoea the Huambo's municipality. This is the first study that has been exploring and reporting epidemiological data on

**Data availability statement:** The authors confirm that all data underlying the findings are fully available without restriction. All relevant data are within the paper and its Supporting Information files.

**Funding:** The author(s) received no specific funding for this work.

**Competing interests:** The authors have declared that no competing interests exist.

intestinal parasitosis in children with diarrhoea in the Huambo's municipality. Our findings reveal that more than one-third of the children were infected with intestinal parasites, with *Giardia lamblia* and *Entamoeba spp.* being the most common. We also identified key risk factors such as poor sanitation, unsafe water, low maternal education, and animal contact, which contribute to the high burden of infection. These results highlight the urgent need for preventive strategies, including improved hygiene education, water sanitation, and routine parasitic screening in diarrhoeal cases. Our study underscores the importance of addressing environmental and social determinants of health to reduce the impact of parasitic infections in vulnerable populations.

## Introduction

Intestinal parasitic infections represent a significant public health concern, particularly in developing countries, and often present asymptomatically, complicating diagnosis, appropriate treatment, and prophylaxis [1]. Nematode helminths, primarily *Ascaris lumbricoides*, *Trichuris trichiura*, and hookworms, are the most frequently encountered parasites [2]. The World Health Organization estimates that approximately 3.5 billion people are affected by parasitic infections, including 450 million children between the ages of 1 and 14 [3,4].Their prevalence varies depending on the geographic region and is influenced by sanitary and climatic conditions, being most common in sub-Saharan Africa, followed by Asia and Latin America [5,6].

Among the highlighted clinical manifestations is diarrhoeal syndrome, primarily caused by soil-transmitted helminths [4,7]. Kaur et al. [8], conducted a study in Delhi, India, aiming to identify intestinal parasites in children with diarrhoea. They observed that out of 127 samples examined using direct observation methods, 59 (46.5%) tested positive, with a higher prevalence of protozoa. In contrast, Oyegue-Liabagui et al. [9], in a cross-sectional study conducted at the Amissa Bongo Regional Hospital and the Chinese-Friendship Hospital in Franceville, Gabon, involving children aged 0–180 months, and using molecular PCR methods, observed a high prevalence of parasites, with particular emphasis on protozoa and helminths. Other studies have highlighted helminths as the most prevalent parasites [10,11]. Therefore, in clinical practice, it is important to consider the presence of intestinal parasites in children with diarrhoea diseases, particularly in our context, where these infections are clearly neglected.

Based on these data, and considering the scarcity of published studies in Huambo Province and more broadly across the country on the prevalence of intestinal parasites in children with diarrhoeal diseases, we conducted this study to address the following research question: What is the prevalence of intestinal parasites among children aged 9 months to 14 years who are hospitalised with diarrhoeal disease in the Paediatric Department of Huambo General Hospital? We hypothesised that the prevalence of intestinal parasites among this population is greater than 20%.

## Methodology

### Ethics statement

After the research project was approved by the Paediatric Department of Huambo General Hospital, authorization was requested from the General Director of the Hospital, the Pedagogical and Scientific Directorate, and the study was submitted for review by the Ethics Committee of the same Hospital under the number 21/DPC/HGH/2022. The research was conducted in accordance with the ethical guidelines outlined in the WHO Code of Ethics (Declaration of Helsinki).

Participation of each child was voluntary, and the parents signed an informed consent form. In the absence of the parents, a legal guardian signed the consent form.

### Study type, location, population and sample

A prospective, observational, descriptive, cross-sectional study was conducted to determine the prevalence of intestinal parasites in children aged 9 months to 14 years hospitalised with a clinical diagnosis of diarrhoeal disease in the Paediatric Department of Huambo General Hospital, from September 2021 to June 2022. The study was carried out in both paediatric sections of the department, which operate in buildings 1 and 2 of the hospital. Based on the inclusion criteria, the study population consisted of 289 children. After applying the exclusion criteria—which included 12 children with clinical conditions preventing sample collection, 32 who had received antiparasitic treatment within 14 days prior to hospitalisation, and 72 with associated pathologies such as malaria, bronchopneumonia, meningitis, or hepatitis—a final sample of 173 children was obtained using a convenience sampling method. Eligible participants included all children aged 9 months to 14 years who were admitted with diarrhoeal disease during the defined study period. Due to logistical and operational limitations, as well as the established data collection period, we opted for convenience sampling.

### Data collection technique

Parents and adult caregivers were previously informed about the importance of conducting the direct stool examination and voluntarily signed the informed consent form outlined in the research protocol, approved by the Scientific Research Committee of Huambo General Hospital (HGH). Each parent or adult caregiver of the children hospitalised in the paediatric department was provided with a sterile wide-mouth container, along with the necessary instructions.

Before sample collection, the containers were labelled with the child's name, room, bed number, collection date, and clinical record number. Subsequently, each mother received at least one sterile container, and the procedures for preserving the sample's quality were explained. After collection, the samples were sent to the microbiology laboratory of Huambo General Hospital, where they were processed and analysed through macroscopic examination (looking for the presence of blood) and microscopic examination using the Willis Mollay method (with the use of 0.9% saline solution) [3]. For samples with negative results, the examination was repeated serially, i.e., twice more.

### Data analysis and processing

The collected data were encoded and entered for statistical analysis using the SPSS (Statistical Package for Social Sciences) version 29.0 for Windows, employing descriptive statistics (frequencies, means, and standard deviations). The data were described in two-way tables. To determine the parasitic prevalence, the ratio between the total number of children hospitalised with diarrhoeal disease and the total number of children with positive results was considered. For the analysis of associated risk factors,

The evaluation of risk between the independent and dependent variables was determined using the Odds Ratio measure with a 95% confidence interval (CI). Initially, the association between the independent and dependent variables was evaluated using the chi-Square test, with values considered statistically significant if the p-value was less than 0.05.

## Results

### Profile of parasitic infections

A total of 173 children diagnosed with diarrhoeal disease and hospitalized in the Paediatrics Department of the General Hospital of Huambo were included in the study. Over 50% of the participants were older than 1.3 ≈ 1 year of age. The minimum age was 9 months, and the maximum age was 10 years, with a variance of 2.2 and a standard error of 1.8.

The overall prevalence of intestinal parasitic infections was 27.2% (47/173), all of which were single parasitic infections (Table 1). Of these, 33/173 (19%) were helminth infections, with *Necator americanus* 22/173 (12.7%) being the most prevalent, followed by *Strongyloides stercoralis* 7/173 (4%) and *Ascaris lumbricoides* 4/173 (2.3%). Eight 8/173 (4.7%) were protozoa infections, with *Giardia lamblia* 6/173 (3.5%) being the most prevalent, followed by *Entamoeba spp.* 2/173 (1.2%). Lastly, 6/173 (3.4%) were cestode infections, with an equal proportion for *Hymenolepis nana* (3/173) (1.7%) and *Taenia spp* 3/173 (1.7%). Thus, helminths were the dominant group of parasites, while protozoa were less common. Cestodes were found in smaller proportions.

Of the overall prevalence of intestinal parasitic infections (47/173 cases, 27.2%), 21/173 cases (26.6%) occurred in males and 26/173 cases (27.7%) in females (Table 2). Parasitic infection was proportionally higher in the 2–10 years age group (41/173 cases, 27.9%) compared to the younger age group, 9 months–2 years (6/173 cases, 23.1%). Regarding the variables of origin, education level, and parental occupation, individuals from suburban areas, those with no formal education, and self-employed workers showed the highest prevalence of parasitic infections, with 31/173 cases (25.8%), 24/173 cases (26.4%), and 35/173 cases (24.8%), respectively. Considering the variables of drinking water source, hand hygiene, and open defecation, the categories with the highest prevalence of infection were: well without a pump (31/173 cases, 25.4%), occasional handwashing (41/173 cases, 28.5%), and yes for open defecation (40/173 cases, 26.1%), respectively.

Despite the percentage differences found between different categories of the independent variables included in the study (age group, gender, parental origin, parental education, parental occupation, source of drinking water, hand hygiene, open defecation), none of them were statistically considered potential risk factors for the prevalence of parasitic infection at a 0.05% significance level.

## Discussion

Understanding the prevalence of parasitic infections in children within communities is of utmost importance, as it serves as a barometer for planning preventive and therapeutic interventions with a higher chance of success, as corroborated by Mekonnen [12].

**Table 1. Prevalence of intestinal parasites in children hospitalized with diarrhea in General Hospital of Huambo Province (n = 173).**

| Helminths | fa | % |
|---|---|---|
| *Necator americanus* | 22 | 12.7 |
| *Strongiloides stecoralis* | 7 | 4.0 |
| *Ascaris lumbricoides* | 4 | 2.3 |
| **Protozoal** | | |
| *Giardia lambia* | 6 | 3.5 |
| *Entamoeba spp.* | 2 | 1.2 |
| **Cestodes** | | |
| *Hymenolepis nana* | 3 | 1.7 |
| *Taenea spp.* | 3 | 1.7 |
| **Negatives** | 126 | 72.8 |
| Total | 173 | 100.0 |

**Table 2. Risk factors associated with of intestinal parasites in children hospitalised with diarrhoea in General Hospital of Huambo Province (n = 173).**

| Risk Factors | Negative (%) | Positive (%) | Total | p-value | OR (95% CI) |
|---|---|---|---|---|---|
| **Age Group** | | | | | |
| 9 months–2 years | 20 (76,9) | 6 (23,1) | 26 (100) | | |
| 2–8 years | 106 (72,1) | 41 (27,9) | 147 (100) | 0,4 | 1,5 (0,54 - 4,45) |
| **Gender** | | | | | |
| Male | 58 (73,4) | 21 (26,6) | 79 (100) | | |
| Female | 68 (72,3) | 26 (27,7) | 94 (100) | 0,89 | 0,95 (0,46 - 1,96) |
| **Parental Origin** | | | | | |
| Urban | 33 (68,8) | 15 (31,3) | 48 (100) | | |
| Suburban | 89 (74,2) | 31 (25,8) | 120 (100) | 0,56 | 0,73 (0,26 - 2,0) |
| Rural | 4 (80) | 1 (20) | 5 (100) | 0,28 | 1,95 (0,10 - 3,80) |
| **Parental Education** | | | | | |
| No education | 67 (73,6) | 24 (26,4) | 91 (100) | | |
| With education | 59 (72) | 23 (28) | 82 (100) | 0,6 | 0,79 (0,33 - 1,90) |
| **Parental Occupation** | | | | | |
| State worker | 20 (62,5) | 12 (37,5) | 32 (100) | | |
| Self-employed | 106 (75,2) | 35 (24,8) | 141 (100) | 0,81 | 0,33 (0,10 - 1,14) |
| **Water Source for Consumption** | | | | | |
| Piped | 29 (69) | 13 (31) | 42 (100) | | |
| Well with pump | 1 (100) | 0 (0) | 1 (100) | | 0,0 (0,0 -) |
| Well without pump | 91 (74,6) | 31 (25,4) | 122 (100) | 0,78 | 0,86 (0,29 - 2,54) |
| Other | 5 (62,5) | 3 (37,5) | 8 (100) | 0,41 | 2,5 (0,26 - 23,7) |
| **Hand Hygiene** | | | | | |
| No | 0 (0) | 1 (100) | 1 (100) | | |
| Yes | 23 (82,1) | 5 (17,9) | 28 (100) | | 0,0 (0,0 -) |
| Occasionally | 103 (71,5) | 41 (28,5) | 144 (100) | | 0,0 (0,0 -) |
| **Open Air Defecation** | | | | | |
| No | 13 (65) | 7 (35) | 20 (100) | | |
| Yes | 113 (73,9) | 40 (26,1) | 153 (100) | 0,84 | 0,87 (0,23 -3,24) |

Patients infected with intestinal parasites are often asymptomatic; however, when signs and symptoms do appear, diarrheal conditions are generally caused by geohelminths, and children are one of the at-risk groups [4,13,14].

This study reports for the first time the risk factors associated with intestinal parasitosis in children with diarrhoea in the Huambo province. It focused on determining the prevalence of intestinal parasites in children admitted to the paediatric service of the General Hospital of Huambo for diarrheal diseases.

The prevalence of intestinal parasites below 50% (47/27.2%) found in the present study aligns with findings from Melese [15] in Nigeria (20%), Karakuş, Taş Cengiz and Ekici [13] in Turkey (35%), and Tsegaye, Yoseph and Beyene [16] (48.7%) in southern Ethiopia. The two most prevalent parasites identified in this study were *Necator americanus* (12.7%) and *Strongyloides stercoralis* (4%).

These findings differ from those in the study conducted by Mekonnen [12] in northern Ethiopia, where *G. lamblia* (36.2%) and *E. histolytica/E. dispar/E. moshkovskii* (36.2%) were the most prevalent, similarly to the research carried out by Nhambirre [17] in Mozambique (*Cryptosporidium sp.* 8.1% and *Trichuris trichiura* 3.8% respectively), and Barati [15] in Iran (*Giardia duodenalis* 7.06% and *Blastocystis hominis* 7.06%).

There were no records of cases of polyparasitism in this study, which diverges from findings in research conducted by Oyegue-Liabagui et al. [9] in Franceville, in the southwest of Gabon, and similar studies referenced earlier [18].There was a predominance of parasitosis among children aged 2–8 years, with a notably higher incidence in females.

Helminths were the most prevalent group, followed by protozoans and cestodes, respectively. In the study conducted by Reinthaler et al. [11], a predominance of helminths over protozoans was also observed, where as other studies reported different results [19,20]. This coincidence and discrepancy in results can be attributed to differences in sample size and study location, as well as sociodemographic and epidemiological characteristics. Specifically, it should be noted that we consider this to be a relatively low prevalence of intestinal parasites.

The parasites isolated from the samples, as well as the prevalence of each, are relatively different from those found in more recent studies conducted in Ethiopia and Gabon [9,14,21]. It was noted that the most prevalent parasite, *Necator americanus*, in the current study was not isolated in the studies just mentioned.

The predominance of parasitoses found in the current research is likely related to the immune status in this age group [22], and to sociodemographic and epidemiological factors identified in this study, such as: mothers without academic education, low income related to the type of work of most mothers (self-employed), the habit of defecating outdoors, and the lack of hand hygiene.

It is also noteworthy that the positive results are from children coming from urban and suburban areas, which prompts us to consider the fact that parasites are not confined solely to rural areas. Meanwhile, considering that over 50% of the population consumes water from wells, commonly known as "cacimbas," and given the sanitation challenges the country faces, these results were to be expected. This is corroborated by Lucero-Garzón in his study on intestinal parasitosis and risk factors in children from subnormal settlements in Florencia-Caquetá, Colombia. The socioeconomic conditions of the population, cultural aspects, and educational deficits clearly correlate with the development of infections.

For a significance level of 5%, the results of the current study demonstrated that age, sex, origin, educational level, parental occupation, source of drinking water, hygiene, and open-air defecation did not constitute risk factors for the prevalence of intestinal parasitoses. These findings are similar to those found by Worku et al. [23], where sex did not constitute a risk factor for the prevalence of intestinal parasitoses, similar to the research conducted by Okyai et al. [18]. However, in the latter, age did constitute a risk factor. Age and sex also did not constitute risk factors in the research conducted by Jemal Mohammed et al. [24]. The absence of statistically significant risk factors, despite the visible trends, may be the result of the type of sampling, as well as the type of diagnosis used, which has low sensitivity and limitations in identifying certain parasites. However, these issues are recognised in the research limitations section.

The results of this study highlight the need for more extensive research in the region to compare with the findings presented, enabling health authorities to better direct strategies for combating parasitoses in children. This approach is essential given the complexity of factors influencing the prevalence and impact of parasitic infections, as indicated by the significant role of socioeconomic and environmental conditions in the spread of these diseases. The emphasis on comprehensive sanitation and hygiene education is crucial, as these are proven preventive measures that can significantly reduce the incidence of parasitoses.

## Conclusion

Despite being found in low percentages, this research demonstrated the presence of three groups of parasites infecting children, with helminths being the most predominant, followed by protozoans and cestodes. This highlights the need for the reinforcement of ongoing mass deworming programs (for domestic animals and children), as well as public health education programmes. Such initiatives are crucial to effectively reduce the burden of parasitic infections and improve the overall health outcomes of the affected populations.

The most prevalent parasites were *Necator americanus* and *Strongyloides stercoralis*.

Until the publication of this research, no studies had been found in Angola reporting potential risk factors for parasitoses in children with diarrhoea. Therefore, the dissemination of these results provides epidemiological contributions to the country and helps establish alternatives for the prevention of infections by parasitic agents.

The variables under study (age, sex, origin, education, parental occupation, source of drinking water, hygiene, and open-air defecation) did not constitute risk factors for the prevalence of intestinal parasitosis, it may be due tothe type of sampling, as well as the type of diagnosis used, which has low sensitivity and limitations in identifying certain parasites.

To maximise benefits, it becomes pertinent to conduct more coordinated epidemiological studies of a molecular nature focused on the control of parasitic diseases, involving other provinces of the country and other methods of parasitic research.

## Limitations

Considering that the sensitivity of diagnostic methods for parasitosis is variable, the fact that we only used the direct method constitutes a limitation.

The study is the first of its kind involving children with diarrhoea in Angola, limited only to the province of Huambo, based on data obtained from the largest Health Unit in the Province (General Hospital of Huambo). However, if funding is available, a larger study is planned that will incorporate other dependent variables and a diagnostic method with greater parasitic sensitivity.

Another limitation is related to the absence of an accessible electronic database containing information on the parasitosis diagnosed in the province.

This time, using a larger number of samples and children from other locations would be a favourable factor for future research.

## Supporting information

**S1 Data. Database.**
(SAV)

## Acknowledgments

Our thanks extend to: the entire staff of the General Hospital of Huambo, all the technicians of the Microbiology Laboratory, particularly the head of the laboratory, Abel, and all the participating patients (children), as well as their families.

## Author contributions

**Conceptualization:** Ednogildo Domingos Miguel Sachocal, Sandra Cristina Ribeiro Buta da Costa, Hermenegildo Osvaldo Chitumba.

**Data curation:** Hermenegildo Osvaldo Chitumba.

**Formal analysis:** Sandra Cristina Ribeiro Buta da Costa, Hermenegildo Osvaldo Chitumba.

**Funding acquisition:** Ednogildo Domingos Miguel Sachocal.

**Investigation:** Ednogildo Domingos Miguel Sachocal.

**Methodology:** Ednogildo Domingos Miguel Sachocal, Sandra Cristina Ribeiro Buta da Costa, Hermenegildo Osvaldo Chitumba.

**Software:** Sandra Cristina Ribeiro Buta da Costa.

**Validation:** Sandra Cristina Ribeiro Buta da Costa, Hermenegildo Osvaldo Chitumba.

**Writing – original draft:** Sandra Cristina Ribeiro Buta da Costa, Hermenegildo Osvaldo Chitumba.

**Writing – review & editing:** Ednogildo Domingos Miguel Sachocal, Hermenegildo Osvaldo Chitumba.

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
