## [Decision Letter · Decision Letter 0]

PNTD-D-24-01894

Prevalence and risk factors associated with intestinal parasitosis in children with diarrhoea in the Huambo´s municipality

Dear Dr. Chitumba,

Thank you for submitting your manuscript to PLOS Neglected Tropical Diseases. After careful consideration, we feel that it has merit but does not fully meet PLOS Neglected Tropical Diseases's publication criteria as it currently stands. Therefore, we invite you to submit a revised version of the manuscript that addresses the points raised during the review process.

Please submit your revised manuscript within 60 days May 16 2025 11:59PM. If you will need more time than this to complete your revisions, please reply to this message or contact the journal office at plosntds@plos.org. Please include the following items when submitting your revised manuscript:

We look forward to receiving your revised manuscript.

Kind regards,

Feng Xue, Ph.D.

Guest Editor

Jong-Yil Chai

Section Editor

Shaden Kamhawi

co-Editor-in-Chief

Paul Brindley

co-Editor-in-Chief

**Journal Requirements:**

At this stage, the following Authors/Authors require contributions: Hermenegildo Osvaldo Chitumba. Please ensure that the full contributions of each author are acknowledged in the "Add/Edit/Remove Authors" section of our submission form.

3) Please upload the main figure as a separate Figure file in .tif or .eps format. For more information about how to convert and format your figure files please see our guidelines: 

https://journals.plos.org/plosntds/s/figures. Please ensure that the figure is labeled as Figure 1.

4) Please provide a complete Data Availability Statement in the submission form, ensuring you include all necessary access information (direct link to access the dataset ) or a reason for why you are unable to make your data freely accessible. If your research concerns only data provided within your submission, please write "All data are in the manuscript and/or supporting information files" as your Data Availability Statement." 

**Comments to the Authors:**

**Please note that one of the reviews is uploaded as an attachment.**

**Reviewers' Comments:**

Reviewer's Responses to Questions

**Key Review Criteria Required for Acceptance?**

**Methods**

-Are the objectives of the study clearly articulated with a clear testable hypothesis stated?

-Is the study design appropriate to address the stated objectives?

-Is the population clearly described and appropriate for the hypothesis being tested?

-Is the sample size sufficient to ensure adequate power to address the hypothesis being tested?

-Were correct statistical analysis used to support conclusions?

-Are there concerns about ethical or regulatory requirements being met?

Reviewer #1: As a result of my examinations made by me, it was concluded that the objectives of the study were clearly stated and an understandable hypothesis was proposed. Considering the study design, it was determined that the planned path was appropriate for realizing the intended objectives.

Even if the sample size selected for the study was not at the desired level, it was considered reasonable, considering the limitations of the study and the fact that they did not receive financial support.

The statistical methods are frequently used in the literature and are preferred in terms of their results.

The authors have no ethical concerns regarding this study. The participants were selected on a voluntary basis, and a consent form was signed. The fact that they also had a valid ethics committee decision on the subject was sufficient to eliminate my concerns.

Reviewer #2: �Are the objectives of the study clearly articulated with a clear testable hypothesis stated?

Yes, the objectives are clear, but a specific hypothesis is not explicitly stated. It would be beneficial to include a testable hypothesis in the introduction.

Is the study design appropriate to address the stated objectives?

Yes, the study design is appropriate for determining prevalence. However, additional justifications regarding sample selection and potential biases would improve the robustness.

Is the population clearly described and appropriate for the hypothesis being tested?

Yes, the population (children aged 9 months to 14 years hospitalized with diarrhea) is clearly described and appropriate for the study objectives.

Is the sample size sufficient to ensure adequate power to address the hypothesis being tested?

The sample size (173 children) seems reasonable for a prevalence study, but the authors do not provide justification for the sample size.

Were correct statistical analyses used to support conclusions?

Yes, the statistical analyses (descriptive statistics, Chi-square test, Odds Ratio) are appropriate for analyzing risk factors. However, some p-values are close to 1, raising concerns about statistical power.

Are there concerns about ethical or regulatory requirements being met?

Ethical approval is mentioned, and informed consent was obtained. No major concerns in this area.

Reviewer #3: No considerations to be applied.

Reviewer #4: Methods

The objective is single, and it is clear along with the research question. Perhaps you could add some specific objectives derived from the identification of genera and species (spectrum of parasitic species and their transmission routes).

Population is clearly described but we recommend to join study location with population and sample in just one paragraph.

The sample size sufficient for statistical power and the selected tested were correct.

The study had the approval of the ethical committee

**Results**

-Does the analysis presented match the analysis plan?

-Are the results clearly and completely presented?

-Are the figures (Tables, Images) of sufficient quality for clarity?

Reviewer #1: The data presented as a result of the analysis conducted within the scope of the research coincided with the plan. The results are clearly and completely presented. Tables and figures were provided to express the results of the research in a comprehensible way and were deemed sufficient.

Reviewer #2: �Does the analysis presented match the analysis plan?

The results align with the methodology, but some statistical findings need clearer interpretation (e.g., why none of the risk factors were statistically significant despite apparent trends).

Are the results clearly and completely presented?

The results are generally well-presented, though some tables and figures (e.g., Figure 1) would benefit from clearer labeling and descriptions.

Are the figures (Tables, Images) of sufficient quality for clarity?

Tables and figures are informative, but figure quality should be improved for readability.

Reviewer #3: No considerations to be applied.

Reviewer #4: The analysis presented match the objectives.

The results are clearly presented, but we suggest in case of Entamoeba histolytica and Taenia solium diagnostics identified just at gender level not species, because by optical microscopy it not possible to asing species level.

Las figuras are in a correct format, but they are not mentioned in the previous text.

In Table 1, we suggest to delete from the last column the information expressed via 1*. It is not required, just de OR of the comparison is correct.

**Conclusions**

-Are the conclusions supported by the data presented?

-Are the limitations of analysis clearly described?

-Do the authors discuss how these data can be helpful to advance our understanding of the topic under study?

-Is public health relevance addressed?

Reviewer #1: The data obtained and presented in this study coincide with data in the literature on the subject. The authors have clearly stated the limitations of the analysis and study, together with the reasons. They have made suggestions for further research on the subject, eliminating the limitations, and continuing with more comprehensive studies.

A fundamental public health issue was addressed when the study was evaluated as a whole. The manuscript addresses the relationship between the subject and public health.

Reviewer #2: �Are the conclusions supported by the data presented?

The conclusions are mostly supported, but the claim that risk factors were not statistically significant could be better contextualized.

Are the limitations of analysis clearly described?

The limitations (e.g., diagnostic method sensitivity, sample size, geographic restriction) are clearly described, but more discussion on potential confounders (e.g., socioeconomic status) would be beneficial.

Do the authors discuss how these data can be helpful to advance our understanding of the topic under study?

Yes, the authors discuss the public health implications and the need for further research and interventions.

Is public health relevance addressed?

Yes, the public health relevance is addressed, particularly in terms of the need for deworming programs and hygiene education.

Reviewer #3: No considerations to be applied.

Reviewer #4: The conclusions are supported by the data presented

The identified clearly the limitations of analysis and make proposals for future research.

This research is valuable, since the results of parasitosis in the area are the first to be reported and would help the Ministry of Health make more targeted interventions for public health.

**Editorial and Data Presentation Modifications?**

Reviewer #1: (No Response)

Reviewer #2: •The authors should consider adding a hypothesis statement in the introduction to clarify the study’s aims.

•The authors should use italics for scientific names

•Provide a stronger justification for the sample size.

•The study did not report any cases of polyparasitism (co-infections with multiple parasites), which is somewhat surprising given the context. This could be due to the limitations of the diagnostic method used.

•Enhance figure clarity with better labels and descriptions.

•While the authors analyzed various risk factors (e.g., age, sex, water source, hygiene practices), none were found to be statistically significant. Clarify why statistical significance was not observed despite apparent trends in the data.

Reviewer #3: No considerations to be applied.

Reviewer #4: The introduction is very brief and I miss more parasitic information from the country of study.

It would be interesting to share in supplementary table form all the information on parasitic identification

**Summary and General Comments**

Reviewer #1: (No Response)

Reviewer #2: (No Response)

Reviewer #3: The study fully meets the scope of the journal and what is expected in scientific research.

Although it is not new to the field, it is relevant because it sheds light on an important public health condition in Angola.

Reviewer #4: (No Response)

PLOS authors have the option to publish the peer review history of their article (what does this mean? ). If published, this will include your full peer review and any attached files.

**Do you want your identity to be public for this peer review?** For information about this choice, including consent withdrawal, please see our Privacy Policy .

Reviewer #1: **Yes: ** Assistant professor Ahmed Galip HALİDİ

Reviewer #2: No

Reviewer #3: No

Reviewer #4: No

**Figure resubmission:**
---

## [Decision Letter · Decision Letter 1]

Dear Professor Chitumba,

We are pleased to inform you that your manuscript 'Prevalence and risk factors associated with intestinal parasitosis in children with diarrhoea in the Huambo municipality' has been provisionally accepted for publication in PLOS Neglected Tropical Diseases.

Best regards,

Feng Xue, Ph.D.

Guest Editor

Jong-Yil Chai

Section Editor

Shaden Kamhawi

co-Editor-in-Chief

Paul Brindley

co-Editor-in-Chief

Reviewer's Responses to Questions

**Key Review Criteria Required for Acceptance?**

**Methods**

-Are the objectives of the study clearly articulated with a clear testable hypothesis stated?

-Is the study design appropriate to address the stated objectives?

-Is the population clearly described and appropriate for the hypothesis being tested?

-Is the sample size sufficient to ensure adequate power to address the hypothesis being tested?

-Were correct statistical analysis used to support conclusions?

-Are there concerns about ethical or regulatory requirements being met?

Reviewer #1: (No Response)

Reviewer #2: (No Response)

Reviewer #3: Done.

**Results**

-Does the analysis presented match the analysis plan?

-Are the results clearly and completely presented?

-Are the figures (Tables, Images) of sufficient quality for clarity?

Reviewer #1: (No Response)

Reviewer #2: (No Response)

Reviewer #3: Done.

**Conclusions**

-Are the conclusions supported by the data presented?

-Are the limitations of analysis clearly described?

-Do the authors discuss how these data can be helpful to advance our understanding of the topic under study?

-Is public health relevance addressed?

Reviewer #1: (No Response)

Reviewer #2: (No Response)

Reviewer #3: Done.

**Editorial and Data Presentation Modifications?**

Reviewer #1: (No Response)

Reviewer #2: (No Response)

Reviewer #3: Done.

**Summary and General Comments**

Reviewer #1: (No Response)

Reviewer #2: (No Response)

Reviewer #3: Done.

PLOS authors have the option to publish the peer review history of their article (what does this mean? ). If published, this will include your full peer review and any attached files.

**Do you want your identity to be public for this peer review?** For information about this choice, including consent withdrawal, please see our Privacy Policy .

Reviewer #1: **Yes: ** Ahmed Galip Halidi

Reviewer #2: No

Reviewer #3: No

---

## [Editor Report · Acceptance letter]

Dear Professor Chitumba,

We are delighted to inform you that your manuscript, "Prevalence and risk factors associated with intestinal parasitosis in children with diarrhoea in the Huambo municipality," has been formally accepted for publication in PLOS Neglected Tropical Diseases.

Best regards,

Shaden Kamhawi

co-Editor-in-Chief

Paul Brindley

co-Editor-in-Chief
